# Associations between Gendered Family Structures and Adolescent Stress, Loneliness, and Sadness in South Korea

**DOI:** 10.3390/ijerph20043656

**Published:** 2023-02-18

**Authors:** Mikaela J. Dufur, Hyeyoung Woo

**Affiliations:** 1Department of Sociology, Brigham Young University, 2008 JFSB, Provo, UT 84602, USA; 2Department of Sociology, Portland State University, 1721 SW Broadway, Portland, OR 97201, USA

**Keywords:** family structure, single parent, single fathers, well-being, mental health, adolescent

## Abstract

While a large literature connects family environments characterized by access to two married biological parents to better child mental health outcomes, we know less about the mechanisms linking family structure to mental health outcomes for children living in other family structures. While essentialist theory suggests that access to both male and female parents will be an important determinant of child mental health, some research directly comparing single-mother and single-father families found no difference in child outcomes by parent gender, suggesting evidence for more structural theories of gender. However, most of this research uses data from Western countries and seldom extends to examining mental health outcomes. In this paper, we used data from a large, generalizable survey of Korean adolescents (the 2021 Korea Youth Risk Behavior Survey) to compare the mental health of children living in families with two married biological parents, single mothers, and single fathers. Our findings underscore the importance of examining family environments in different contexts.

## 1. Introduction

A robust literature connects intact family environments, or family environments characterized by two biological parents who are married to each other, with a wide variety of positive child outcomes including academic success, lower levels of behavior problems, and better health [1]. However, findings from such research have popularly been used to conclude that fathers and mothers contribute separate, unique resources to their children’s development, and that therefore children in single-mother or single-father environments lack access to important, gendered resources. These conclusions, in turn, seem to bolster claims about immutable gender differences within and beyond families [1]. Research that directly compares single-mother and single-father contexts does not support such gender essentialist conclusions [2,3,4]. However, such research has been derived almost exclusively from data on families in Western contexts and from datasets that have relatively few single-father environments (usually fewer than 500) [2,3,4]. In this paper, we extend the examination of single-mother and single-father family environments to South Korea (hereafter Korea), an important extension because custody law and tradition in Korea more often assigns child custody to men after separation, creating more single-father families than in Western contexts. In addition, persistent and strict conservative gender ideologies in Korea provide an ideal context in which to examine how parental gender influences child outcomes, in this case, mental health, as expressed by levels of stress and feelings of loneliness and sadness.

### 1.1. Are Fathering and Mothering Different?

Although a large body of research has demonstrated that mothers do the most parenting work [5], recent research has shown a trend of the growing contributions of fathers to parenting. For example, even though tendencies for high-intensity parenting have led to an increase in maternal parenting time over the past few decades, the ratio of maternal to paternal parenting time has decreased in several Western countries over the same period [6,7]. There are similar patterns, if not as pronounced, in the Korean data, with fathers spending slightly more time on childcare and housework in 2014 than they did in 1999; during the same period, the childcare time of Korean mothers (already higher) increased at about the same rate while their housework time dropped slightly [8]. Women and men spend their parenting time in different ways, however, with women providing more daily necessities (e.g., clothing, feeding, changing diapers) and men being more likely to engage children in play; this is true in both the U.S. and Korean samples [5,6,7,8,9,10].

These differences in mothering and fathering have been identified through examining two-parent families, but little is known about gendered differences in parenting in single-sex contexts. In this paper, we focused on two comparisons: looking at how single-mother and single-father family environments are associated with child mental health compared to two-parent families, and examining single-parent family environments to see whether there are differences in child mental health in single-mother and single-father families in Korea.

### 1.2. Competing Perspectives on Sources of Gendered Differences in Parenting

Although several theories have been proposed to explain why living in single-parent families is associated with poorer outcomes of the children, differences in the outcomes between children in single-mother and single-father families have not been as well explored. Nevertheless, two theories have guided us to explain general differences among youth who lack access to either a mother or a father: essentialist and constructivist theories [2,3,4]. The essentialist perspective suggests that biological mechanisms or early socialization lead to fundamental differences between the sexes, causing women and men to behave–and to be–different [11,12]. According to this essentialist perspective, mothering and fathering will be different, explicitly because women do mothering and men do fathering. Indeed, much work on fathering demonstrates that fathers do parent, but that they do so in different ways than mothers, making mothering and fathering two separate components of parenting [9,13,14,15]. While it may seem surprising that essentialist theory continues to hold sway in highly-educated and industrialized countries, proponents of essentialist theory continue to drive conservative policies and messaging [11].

In contrast, the constructivist perspective suggests that gender is not a set of immutable traits linked inextricably to biological sex; rather, women and men face different structural constraints and expectations in society that shape their behaviors (for foundations of constructivist gender theory, cf. [16,17,18]). Some support for the constructivist perspective comes from meta-analyses that showed few differences between men and women across a number of psychological dimensions, even as men and women experience substantially different life outcomes [19]. According to this perspective, men and women behave differently because of frequent opportunities and strong expectations to ‘‘do’’ gender, or act out socially constructed gendered scripts [18]. The constructivist perspective connects differences in mothering and fathering to broader societal norms about acceptable gendered behaviors, both within and beyond families [6,7]. In contrast to the essentialist position, the constructivist perspective predicts that single mothers and single fathers will perform similar parenting functions simply because they must—that mothers and fathers will be equally concerned with providing the resources all children need (e.g., food, shelter, and clothing; education; emotional and financial support; discipline) but will lack an opposite-gender partner with whom to act out a gendered script stating who should provide specific resources [3].

These two theories offer contrasting predictions, with the essentialist theory stating that experiences will differ for children in single-mother vs. single-father families and the constructivist theory stating that youth in both family contexts will have similar experiences. At the same time, both of these theories concerning gender leave open the possibility that children in either single-mother or single-father families would do worse compared to those in intact families because having two parents available brings with it a number of socioeconomic and social resources [2,3,4].

### 1.3. Cross-Cultural Findings on Parenting in Single-Parent Contexts

To date, we know little about whether single-parent family environments outside of the U.S. reflect similar gendered patterns, and what those patterns might teach us about gendered underpinnings of family life. Recent research looking at single-father and single-mother families in the United Kingdom found few differences between the family structures when predicting child behavior problems, mirroring the U.S. findings and further suggesting that parents move beyond gendered behaviors when solely responsible for caretaking [20].

The East Asian setting is a particularly interesting one to use to examine these questions because it provides opportunities to study a context in which family life is in flux. On one hand, patterns of family formation and disruption in non-Western nations occur in distinct historical, social, and cultural contexts. Historically, divorce, cohabitation, and single parenthood have been much less common in East Asian nations than in Western nations [21,22,23]. However, in high-income Asian nations like Korea, a recent second demographic transition has influenced family life, family structure, and child outcomes in a uniquely Korean manner [24]. Over the last 30 years, Korea has seen an increase in crude divorce rates as part of a second demographic transition, from 1.1 divorces per 1000 people in 1990 to 2.1 divorces per 1000 in 2017 [25]. However, the cultural pressures to maintain a traditional family formation sequence (i.e., get married and have children) and structure—with two married parents and a male breadwinner—have remained ingrained in Korean society [26], leading some scholars to conclude that the Korean context is one where family environments are subject to both rapid changes and to strong traditional and gendered pressures [23,24,26].

On the other hand, there is some evidence for similar relationships between family environments and child outcomes in the East Asian context, as exist in Western contexts. Research in Hong Kong, Korea, and Japan found that children raised in single-parent families had lower academic achievement than children in two-parent families and that socioeconomic disadvantage is a key factor in this disparity [27,28,29], similar to findings from Western samples. What is unique to the East Asian context is that the achievement gaps based on family structure are smaller than in the U.S. [30]. These differences and similarities underscore the importance of comparative work on family environments for understanding how well-supported broad theories about family are.

Pertinent to our question here about potential gender differences in family contexts, evidence from East Asian samples concerning academic outcomes suggests some striking differences across family environments that are in marked contrast to the findings in the U.S. In a sample of adolescents from Hong Kong, students from single-father families scored worse on standardized tests than those in single-mother families, largely due to single fathers having fewer socioeconomic and social resources [28]. In a Japanese sample, single fathers were similarly less involved at home and school, but gendered labor dynamics translated to fewer socioeconomic resources for children in single-mother homes [31]. In a sample of Chinese adolescents, children from single-father families were in bivariate relationships that were not statistically significantly different from those in intact families, while children in single-mother families did better in terms of self-evaluated academic performance and performance on standardized tests. After controlling for a number of family environment factors, however, children in single-mother families performed about the same as those in intact families, while those in single-father families did worse [32]. Findings from each of these East Asian samples highlight what might be thought of as deficiencies in single-father families including difficulties in creating the social resources that might be associated with mothers or women. These patterns are in sharp contrast to the U.S. samples, where adolescents in single-father families tend to do slightly better than those in single-mother families until income is controlled [2,3,4], and children tend to be exposed to equally involved parents regardless of parental sex [33,34,35]. Findings like these suggest that the conclusions from the U.S. data concerning how gendered parenting operates may not be universal, highlighting the need to examine cases beyond the Western context. In this line of inquiry, Korea’s persistently conservative gender roles provide a useful contrast to the Western setting [26].

### 1.4. The Current Study

Our study extends past research by looking outside the U.S. context and by examining the children’s mental health rather than their academic performance. Findings from the U.S. studies suggest that once we include the controls for socioeconomic resources, adolescents in single-mother and single-father families will not differ much in terms of their mental health, which would provide support for the constructivist position on gender. However, the work on academic outcomes in East Asian samples suggests a perhaps more complicated set of mechanisms, and the possibility of support for the essentialist position on gender in families.

The current study set up two tests: first, we examined the potential differences in child mental health outcomes among children in two-married-biological parent families and those in either single-mother and single-father families to establish a baseline comparison to work in the U.S. and other Western contexts that found negative associations for children in single-parent families and to work in Korea that found negative associations between academic outcomes and living with a single mother or single father. We then turned to a comparison between adolescents in single-mother and single-father families to see whether children in Korea exhibited patterns similar to those in the U.S., where small differences across family context are largely explained away by single fathers’ greater access to physical or financial resources, or whether the findings on child mental health will more closely mirror work on single-mother and single-father families and academic outcomes in other East Asian contexts, where more complicated patterns emerge.

## 2. Methods

### 2.1. Procedures and Participants

Data came from the 2021 Korea Youth Risk Behavior Survey (hereafter KYRBS). The KYRBS has been conducted by the Korean Disease Control and Prevention Agency (KDPCA) every year since 2005 based on a national sample of schools to collect information about the health conditions, health behaviors, and family background among children and adolescents in Korea. The KDPCA IRB provides ethical approvals and oversight for the survey. The KDPCA obtains consent from parents and students, and teachers who go through KDPCA survey training administer the survey in the sampled schools. Among 17 surveys of the KYRBS that have been collected, we used the 2021 survey, which is the most recent one currently available. The KYRBS offers several unique advantages for this research. First, the KYRBS offers information on various health outcomes including mental health measures (e.g., stress level, loneliness, and sadness) and general health (e.g., self-rated health) as well as family structure. Second, as a school-based survey, students in a sample of middle and high schools across the country (7th to 12th graders) take the survey while at school, and thus the response rate is nearly 100%. Third, the 2012 KYRBS contains a large number of adolescent respondents, with more than 54,000 adolescents surveyed, allowing us to produce reliable estimates of the mental health of adolescents who are in a single parent homes in comparison with those who are in two-parent homes. In doing so, we limited our analysis to these three groups (i.e., adolescents living with two biological parents, adolescents living with a single father, and adolescents living with a single mother) excluding cases in which another adult was in the household to avoid complexities and variations in parental involvements and relationships with children from step-parent families or grandparent families. After excluding respondents who lived with anyone other than “both parents”, “single father”, and “single mother”, the analytic sample included 38,383 respondents including those with both parents (*n* = 34,735), single father (*n* = 897), and single mother (*n* = 2751). We used SAS 9.4 to analyze the data for this study.

### 2.2. Dependent Variables

To estimate the mental health of adolescents associated with family structure, we used three mental health outcomes: levels of perceived stress, loneliness, and sadness. For the levels of perceived stress, we used a question asking “How much do you usually feel stressed?” on a 5-point scale ranging from 1 (“not at all”) to 5 (“very much”). The variable for loneliness is measured in a similar way by a question that asks, “How often have you felt lonely in the past 12 months?” The students’ responses were coded 1 for “hardly” through 5 for “always.” As for sadness, the KYRBS asked whether the students had felt extremely sad or frustrated in the past 12 months, and we coded 0 for “no” and 1 for “yes.”

### 2.3. Primary Independent Variable

With respect to family structure, the KYRBS provides information about household rosters. Given the information, we categorized students into three groups to represent the types of family structure in which students reside: two biological parents; a biological single father; and a biological single mother. As mentioned earlier, we excluded students who lived with step-parent(s) or non-parent guardian(s) to ensure that our estimates captured the differences between two-parent families and single-parent families. Ideally, we would also include measures of parent gender expression and gendered behavior to examine how gendered family structures might affect child outcomes. Unfortunately, the KYRBS does not contain any such parental variables; as a result, in keeping with the literature that makes similar single-mother to single-father comparisons [2,3,4], we used the sex of the parent in single-parent family as a proxy for gender.

### 2.4. Control Variables

To mitigate potential selection effects, we included several variables to account for a student’s demographic information, family background, student’s school performance, perceived body image, self-rated health, and locality. In addition to being associated with child behavioral and health outcomes [36,37,38,39,40], previous research using Western data has shown that these factors typically differ across not only two-parent and single-parent families, but across single-father and single-mother families [2,3,4]. Specifically, for the demographic information and family background, we included the student’s grade (7^th^ to 12^th^), sex (0 = “boy” vs. 1 = “girl”), family’s socioeconomic status (1 = ”low” to 5 = ”high”), and father and mother’s education (“high school or less”; “some college or more”; and “don’t know”). For a student’s self-assessment on school performance, body image, and general health condition, we used self-reported school performance (1 = ”low performing” to 5 = ”high performing”), body image (“too skinny”, “normal”, and “too fat”), and self-rated health (1 = ”poor” to 5 = ”excellent”). Finally, we included where students resided (“rural area”, “small city”, and “big city”). See Table 1 for the additional measurement details for all variables.

### 2.5. Analytical Plan

We first present the descriptive findings for the total sample and then across all three family types we examined here (living with both parents, living with a single father, and living with a single mother). We then tested our baseline idea that adolescents living with a single parent in Korea will report worse mental health than their peers living with both parents by presenting multivariate regression models that predict child stress, feeling lonely, and feeling sad. We then tested whether adolescents in single-father and single-mother contexts reported different mental health by repeating these multivariate analyses only on the sample of adolescents who lived with a single parent. Finally, to explore potential variations by gender, we also included an interaction between family structure and gender in the models. Models with a dichotomous dependent variable (whether respondents have felt extremely sad or frustrated in the past 12 months) employed logistic regression models; models predicting adolescent stress and how often the adolescents felt lonely in the past 12 months employed ordinary least squares (OLS) regression. All of the regression estimates were weighted to account for the school-based sampling design.

## 3. Findings

### 3.1. Distribution of Family Structures

Table 1 shows the description for all variables in the model for the total sample and for adolescents in each family structure (both parents, single father, and single mother). The average score of stress level was 3.27 on a five-point stress scale (1 = not at all to 5 = very much); adolescents from both single-mother and single-father families scored slightly higher than those living with both parents. Adolescents from both forms of single-parent families also scored higher than those living with both parents on how lonely they felt in the past 12 months or whether they had felt sad in the past 12 months; the largest gap was between adolescents in single-father families and two-parent families reporting whether they had felt sad. Adolescents in both types of single-parent families reported performing worse in school on average and having lower socioeconomic status. Both patterns are similar to those in Western samples, and, regarding the academic outcomes, work from East Asian samples [29,30]. Adolescents in single-father families were exposed to mothers with more education than children in the other family types; this pattern was repeated for fathers of children in single-mother families. Scores for perceived body image were similar across all three family groups, though more adolescents in single-father families reported feeling “too fat” compared to those in two-parent families. There were few notable differences in locality across the family contexts.

### 3.2. Child Stress and Living with One Versus Two Parents

We now turn to the regression models designed to test our baseline question, whether adolescents in both single-father and single-mother families would report worse mental health in the form of stress, sadness, and loneliness than adolescents who lived with both parents. Table 2 presents the regression models predicting adolescent perceived levels of stress. Model 1 shows the bivariate relationships between the family structure and stress. Adolescents in both single-mother and single-father families scored significantly higher on the adolescent stress instrument than those living with both parents, though the coefficients were small. When adding controls for adolescent sex, grade in school, family SES, and parental education, differences between adolescents in single-parent families and those living with both parents decreased but remained statistically significant (Model 2). Model 3 adds controls for adolescent school performance, body image and rurality; again, these attenuated some of the effects of living in a single-parent family compared to living with both parents, especially for single-father families. While coefficients for living with a single mother or single father remained significant, they were at this point quite small. Model 4 introduces the interaction effects between living in single-parent family structures and adolescent sex. Both of these interactions were significant; Figure 1 shows these effects, which demonstrate that girls living with single mothers reported the highest levels of stress. These findings provide support for our baseline test that living with either kind of single parent is associated with greater stress than living with both parents, though these effects seem to be smaller than those often found in Western samples. In addition, these findings are suggestive of interesting gender effects that require more exploration comparing single-mother and single-father families, which we return to below. The control variables acted as we expected; for example, adolescents with access to more financial and human capital reported less stress, while those with poor body image reported more stress.

### 3.3. Child Loneliness and Living with One Versus Two Parents

Table 3 used OLS regression to predict student scores on how lonely adolescents reported they felt during the past 12 months (higher scores = greater loneliness). As was true for models predicting child stress, living with either a single mother or a single father was associated with higher loneliness scores compared to living with both parents (Table 3, Model 1). This remained true even after controlling for child sex, grade in school, family SES, and parental education (Model 2), and for child school performance, body image, and locality (Model 3). As was true for models predicting child stress, the control variables behaved in expected ways. Model 4 introduced interaction effects between those living in single-parent family structures and child sex. These interactions were statistically significant; Figure 2 shows a sharp effect of being a girl on higher reported loneliness, especially for girls in single-father families. This pattern was different from the findings for feeling stress, where girls in single-mother families scored higher. Similar to the findings concerning child stress, these results provide support for our baseline test that living with either kind of single parent was associated with a greater likelihood of feeling lonely than living with both parents, but these findings introduce additional questions about the potential differences between living with a single father or a single mother.

### 3.4. Child Sadness and Living with One Versus Two Parents

Table 4 repeats these analyses using logistic regression models to predict the odds of having felt sad in the last 12 months. Again, Model 1 presents the bivariate analysis including only family structure context. Adolescents in both single-father and single-mother families reported significantly higher odds of having felt sad than those who lived with both parents. These patterns persist when including controls for child sex, grade in school, family SES, and parental education (Model 2). Model 3 adds controls for school performance, body image, and locality; adolescents in both kinds of single-parent families remained more likely to report having felt sad in the past 12 months. The control variables operated in the expected ways and similarly to how they operated in the models predicting stress and loneliness. Model 4 introduced interaction effects between family structure and child sex; these interaction effects indicated that girls in single-father families reported higher odds of having been sad (Figure 3). These findings are more similar to those for child loneliness than for child stress, where girls living with single mothers reported higher scores. In addition, these interaction patterns predicting the odds of being sad were more similar to those from models predicting loneliness; while girls reported higher scores or odds across all three family structures on all three outcomes, the effect of child gender was stronger for loneliness and sadness than for stress, perhaps because the former was measured in the past 12 months. Similar to findings concerning child stress and loneliness, these findings provide support for our baseline test that living with either kind of single parent is associated with a greater likelihood of feeling sad than living with both parents, though the effects for sadness were slightly less sensitive to the inclusion of controls.

### 3.5. Child Stress and Living with a Single Mother Versus a Single Father

With the baseline finding that adolescents in both kinds of single-parent family reported more feelings of stress, sadness, and loneliness established, we then moved on to tests regarding our second research question, whether those who lived with a single father or a single mother reported different mental health outcomes. These analyses only included adolescents who lived with a single parent; those living with single fathers were the reference group. Table 5 presents the OLS regression models predicting child stress. In the bivariate (Model 1), adolescents living with single mothers reported significantly lower stress levels than adolescents living with single fathers. This difference was very small; however, it increased slightly with the inclusion of variables measuring child sex, grade in school and parental SES and education (Model 2). However, the difference in stress between adolescents in single-father and single-mother households decreased to close to the bivariate coefficient with the inclusion of variables of school performance, body image, and locality; still, this small difference remained statistically significant, indicating greater protective effects of living with a single mother. Interactions between child sex and family type indicate that girls in single-father families reported the highest stress levels, while the stress levels of boys in either family type were very similar to each other (Figure 4).

### 3.6. Child Loneliness and Living with a Single Mother Versus a Single Father

Table 6 used OLS regression models to examine feeling lonely in the past 12 months. The results were similar to the models predicting stress, with small advantages for adolescents living with single mothers compared to their peers in single-father families (Model 1). Although this negative association persisted as significant in subsequent models adding controls (Models 2 and 3), the results for feeling lonely differed slightly in that the addition of each theoretical block explained more of the difference in loneliness between adolescents with single fathers and those with single mothers. Like the results for stress, girls uniformly reported higher stress than boys, with girls in single-father families reporting the highest levels of loneliness (Figure 5). Boys in single-father families also reported more loneliness than their same-sex peers in single-mother families, perhaps reflecting that fathers are less embedded in social networks themselves.

### 3.7. Child Sadness and Living with a Single Mother Versus a Single Father 

Finally, Table 7 extended this exploration by using logistic regression models to examine the odds of having felt sad in the past 12 months. As was true for the models predicting stress, living in a single-mother family was negatively and significantly associated with feeling sad (Model 1). This significant relationship persisted through subsequent models introducing child sex, grade in school, parental SES, and education (Model 2) and school performance, body image, and locality (Model 3). The interaction effects between adolescent sex and parent sex were in some ways similar to those we saw when predicting stress; girls in single-father families reported the greatest likelihood of having felt sad, and higher odds than their female peers in single-mother families. However, unlike the patterns for loneliness, boys in single-mother families reported being more likely to have felt sad than their same-sex peers in single-father families (Figure 6).

## 4. Discussion

We set out to first examine whether patterns of child outcomes in single-father and single-mother families would indicate deficits in child outcomes compared to two-parent families in Korea. We then expanded this inquiry to whether comparisons of child mental health, measured as stress, loneliness, and sadness, across single-mother and single-father families in Korea would provide support for gender constructivist theories, as we found in samples from Western countries, or gender essentialist theories, which have been suggested by some research in East Asian countries that found deficits primarily for adolescents in single-father families. Comparisons to two-parent families demonstrated patterns very similar to those in Western countries—adolescents in single-parent families led by either mothers or fathers lack key resources compared to those in two-parent families that leave students in single-parent families at higher risk compared to those who live with both a mother and a father. Our findings also extend these patterns beyond the academic outcomes already examined in East Asian settings to important mental health outcomes. Similar to research on data from Western countries, these initial findings seem to provide evidence for essentialist perspectives.

However, in the Western data, models that control for additional family characteristics and place single-mother and single-father families in similar demographic and economic contexts demonstrate support for constructivist theories, showing that both parents and children in single-mother and single-father families are similar. For the few variables on which differences do exist, Western samples usually suggest small benefits for living with a single father, and these benefits often disappear when single-mother families occupy similar structural positions as single-father families in terms of having more access to financial and human capital [2,3,4]. We did not find these patterns when directly comparing children in single-mother families to those in single-father families in Korea. Results from this Korean sample differ from samples in the West in that they revealed a consistent, if sometimes small, advantage for adolescents living with single mothers compared to those living with single fathers, even in the presence of controls. These complex patterns underscore the importance of examining family environments and mechanisms as well as gender across different settings.

Why might gender operate differently in single-parent families in Western and Korean contexts? One possibility is the persistence of conservative gender norms in Korea [26]. While there have been slight increases in the time fathers spend on child care and housework in Korea, these increases are much smaller than those reported in Western samples, and the gap between maternal and paternal time in such endeavors remains consistent in Korea, while shrinking in Western contexts [6,7,8]. Single fathers in Korea may not have participated as much in child care before becoming single parents as those in Western countries and therefore may need a longer adjustment period before viewing themselves as caretakers. Korean single fathers may also not be as well-networked into parent networks that can provide them with support and help. Our findings for child loneliness are suggestive of the idea that single fathers are not as well-networked. These possibilities provide intriguing puzzles for trying to use data on single-mother and single-father families to distinguish whether essentialist or constructivist theories of gender better explain family life: our findings could be taken as evidence that mothers are superior parents even in difficult and resource-poor contexts than fathers, a position supported by proponents of essentialist gender theories. However, explanations for our findings suggest that Korean single fathers are products of conservative gender norms and that they could grow into their roles, offering a different kind of support for constructivist gender theories.

Similarly, analyses using Western data that compare boys and girls living with single fathers to those living with single mothers have found very little evidence of a moderating effect of child or parent gender on child outcomes [4]. In these Korean data, however, gender appeared to be more salient. Girls reported higher levels of mental health concerns, which we might expect given persistent cross-national patterns suggesting girls and women might be more comfortable expressing emotion; this may remain especially true in Korea, where strong and conservative gender norms still exist [26,41]. When considering parent and child gender in tandem, however, the findings in these Korean data are quite different than their Western counterparts. Girls in single-father families do significantly worse in terms of mental health than their peers in single-mother families; this is true for boys only when considering loneliness. Not only does living with a single mother show significant, if small, protective associations for the mental health of adolescents, this is especially true for girls. It is possible that girls are more attuned to social structures and their place within them, especially as pertains to gendered structures in a society with persistently conservative gender norms [26]. Given that living in a single-parent home at all is non-normative in Korean society, and that living in a single-father home is less common than living in a single-mother home, girls living with single fathers may be especially sensitive to social pressures regarding nonconformity.

Conservative gender norms in Korea might also lead families to invest more resources in sons than in daughters [26,42]. Resource patterns among single-parent families in Korea are quite different than those in Western contexts; in both contexts, single-parent families have fewer resources than two-parent families. In the Korean data we used here, however, single-mother households enjoyed better socioeconomic positions than single-father households, the opposite of the findings in Western data. If daughters are already receiving lower investments from their families, daughters in single-father families might be especially at risk of insufficient investment, and as a result, worse mental health outcomes. Similarly, because of the persistent conservative gender norms in Korea, single mothers in this context may feel greater pressure to “prove” themselves by investing social resources into parenting. Single fathers, on the other hand, might feel greater pressure to prove themselves in the workplace. This might create situations where they are not only less able to invest social resources in their children, but perhaps especially in their daughters. Moreover, if they are unsuccessful in such endeavors, as might be indicated by their lower socioeconomic positions in these data, their own mental health might also suffer, making them less able to play a parent role for their children. Data from other East Asian contexts suggesting that both parents and children “double down” on gender stereotypes in single-parent families provide some support for this explanation and point out connections to constructivist theories of gender, since gender behaviors are fluid in these settings [28]. Single-mother families in our sample are also more likely to have daughters, which may indicate a selection effect of some kind that could be related to single fathers being less likely to choose to parent daughters (or daughters may choose to live with the mothers), or a gendered assumption on the part of family courts on whether daughters are better off with their mothers. Again, the initial findings here seem to indicate support for essentialist theories of gender, with men appearing less ideal for parenting girls. However, if explanations for these patterns are rooted in societal gender ideologies, this would instead be indicative of support for constructivist gender theories.

Our paper had several limitations. The data did not include questions about the parents’ personal gender ideologies, so we used sex as a proxy for gender. Being able to directly test the parents’ gendered attitudes or behaviors would be useful. While variables asking about loneliness and sadness asked about the past 12 months, the question on stress contained no such time boundaries, which may help to explain the smaller effects for that outcome. The data lacked information on time-sensitive potential covariates such as the children’s own peer/romantic relationship status or school exams, which could be important causes of stress, loneliness, or sadness. In these cross-sectional analyses, we could not tell how long the parents and children had been in a single-parent family, how they got there, how much contact the children had with their non-residential parents, or how healthy or competent those non-residential parents were.

We were also only able to measure our mental health outcomes with single indicators for each concept; future research that can examine richer and broader measures of these concepts will be useful. The cross-sectional nature of the analyses also meant that we could not test the explanations that we proposed above about whether single fathers did less caretaking before becoming single or were less networked compared to single mothers; such analyses would allow for further evidence in favor of constructivist gender explanations, but are beyond the scope of these data. Future research examining longitudinal data could help untangle such questions. Additional opportunities for future research include extending our questions about non-traditional family structures and gender theories to other family contexts such as step-parented families or families with a grandparent as a primary care provider for the children to see how gender plays out in other family structures. Though beyond the scope of this study, the approaches we present here could also be used to study how siblings mediate or moderate the relationships between single-mother or single-father status and mental health. On one hand, it is possible that siblings could help bring resources into the home that would benefit the mental health of adolescents; on the other hand, it is possible that the presence of siblings could further dilute resources in single-parent families that are already at a deficit compared to two-married-parent families. In order to fully understand the potential sibling effects, it would also be important to understand where the respondent child falls in the birth order, as we would expect younger siblings to require more time for direct supervision and older siblings to take up more resources as well as for older siblings to be expected to take on more child care duties or household chores in single-parent families. In addition, the gender composition of sibships could further complicate the ways of living with a single-father or a single-mother, which might affect adolescent mental health. As an example, perhaps a girl living with both a single-father and brothers might suffer even worse mental health effects because of the gendered practices in both her own home and in society. While the KYRBS data we used here could potentially be used to answer such questions, it is important to note that low fertility in Korea means that many adolescent respondents would not have siblings [43]. Questions about sibling effects might need to be extended to other settings with higher fertility to be fully explored. It would also be useful to see patterns of remarriage among single-parent groups in order to better examine ideas about resource deficits, helping to explain the single-father effects we found here. Single fathers with serious financial or social deficits may also be less likely to remarry.

Finally, we note that our work here is consistent with a tradition of examining family structure and associated outcomes for children only. While this is a common approach because scholars and policymakers alike are extremely concerned about how to encourage prosocial child development, it is also an approach that ignores the family as a system or how family structures affect parents, at least in terms of outcomes. For example, while parents in Scandinavian countries on average represent contexts that are both above (Norway, Sweden) and below (Denmark, Finland) the international averages for raising children in traditional marriages than in many other parts of the world [44], they also report greater happiness [45] and greater resilience in the face of stressors [46]. At the same time, the negative effects of living with a single-parent on academic outcomes were also average compared to the rate of single-parenthood in these countries [44]. Most pertinent to the questions we present here, we know very little about what comparing single-mother and single-father families in these contexts can teach us about gendered family mechanisms or gender theory in general. Readers should also exercise appropriate caution in how to generalize or use these findings, especially in public or policy spaces. We note that our analyses here are meant to adjudicate between large-scale theories about gender and to underscore the importance of looking at family mechanisms across many cultures, but not to make claims about how individuals should construct their personal families or parent their own children.

## 5. Conclusions

In the end, our findings contribute to a better understanding of the complex picture of how gender operates within families. Comparing single-mother and single-father families allows for a more direct test of whether gendered behaviors are inborn traits or socially constructed, but divergent findings in Asian and Western contexts suggest a need for further exploration. Using the models we constructed here in other societies with very intensive parenting norms at home and strong gendered expectations in society might provide further insights into this question. Still, the very fact that there are differences across contexts suggests support for the constructivist perspective—if behaviors and outcomes in single-parent families are different from two-parent families in different societies, it seems unlikely that gendered parenting behaviors are inborn, essential traits already present within parents and unshaped by social constructs. Educational and medical personnel must be aware of both the difficulties that children in single-parent families face, but also of the specific nuances concerning gender and family structure that operate in their settings and that might affect the well-being of the children with whom they work. Similarly, policymakers should be encouraged not only to pursue approaches that make up resource deficits for non-traditional families, but to understand the gendered specifics of family dynamics in their contexts.

## Figures and Tables

**Figure 1 ijerph-20-03656-f001:**
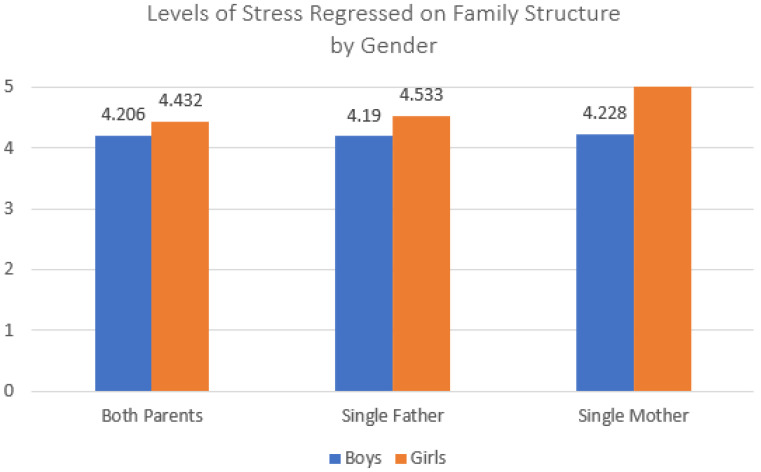
Interaction effect between family structure and child sex predicting child stress. Note: Data source is the 2021 Korea Youth Risk Behavior Survey; total *n* = 38,383 respondents (both parents *n* = 34,735; single father *n* = 897; single mother *n* = 2751).

**Figure 2 ijerph-20-03656-f002:**
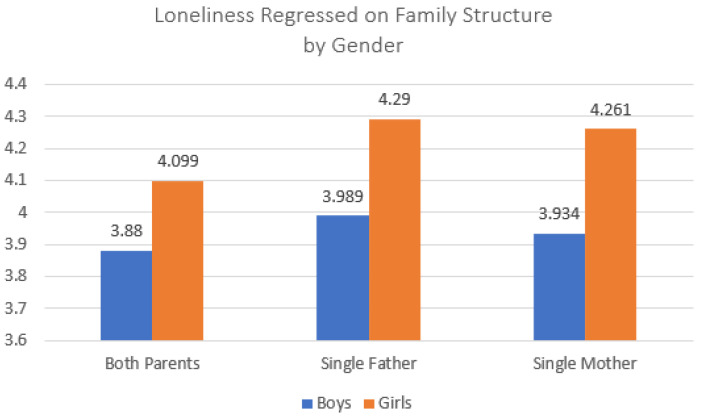
Interaction effect between family structure and child sex predicting child loneliness. Note: Data source is the 2021 Korea Youth Risk Behavior Survey; total *n* = 38,383 respondents (both parents *n* = 34,735; single father *n* = 897; single mother *n* = 2751).

**Figure 3 ijerph-20-03656-f003:**
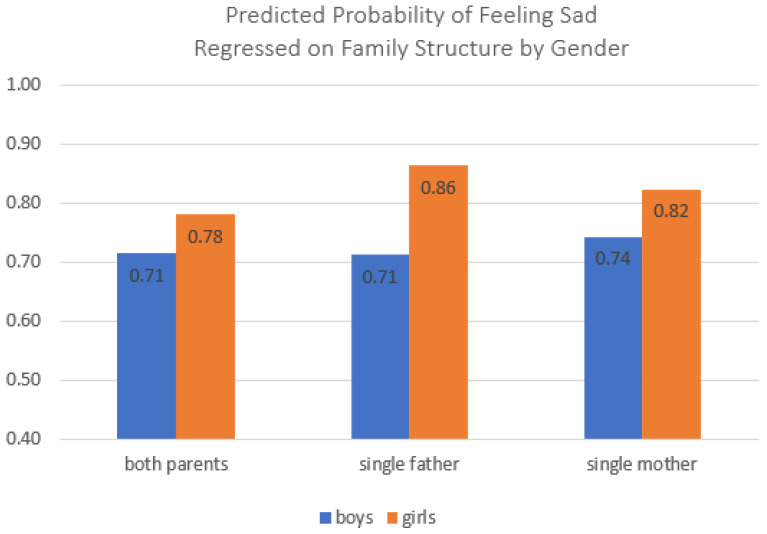
Interaction effect between family structure and child sex predicting child sadness in the past 12 months. Note: Data source is the 2021 Korea Youth Risk Behavior Survey; total *n* = 38,383 respondents (both parents *n* = 34,735; single father *n* = 897; single mother *n* = 2751).

**Figure 4 ijerph-20-03656-f004:**
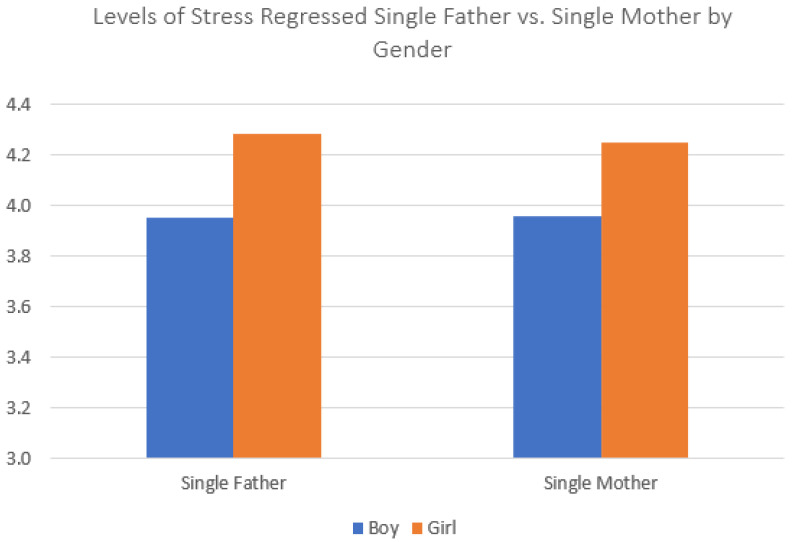
Interaction effect between single-father/single-mother status and child sex predicting child stress. Note: Data source is the 2021 Korea Youth Risk Behavior Survey. Single father *n* = 897; single mother *n* = 2751.

**Figure 5 ijerph-20-03656-f005:**
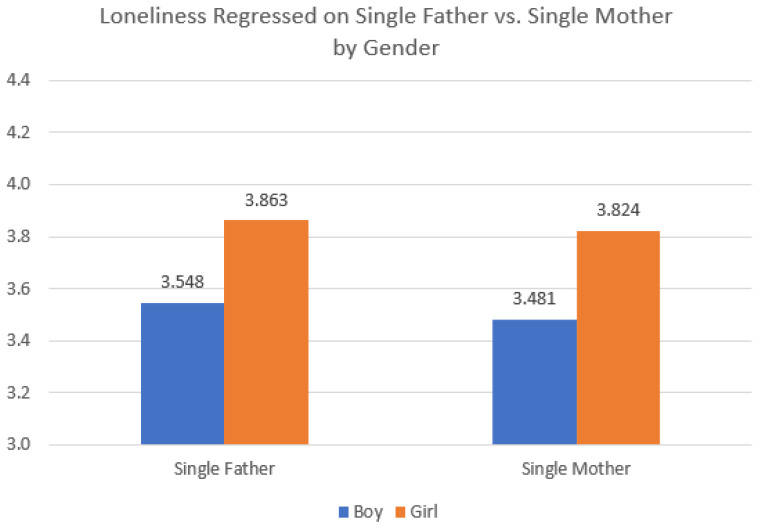
Interaction effect between single-father/single-mother status and child sex predicting child loneliness. Note: Data source is the 2021 Korea Youth Risk Behavior Survey. Single father *n* = 897; single mother *n* = 2751.

**Figure 6 ijerph-20-03656-f006:**
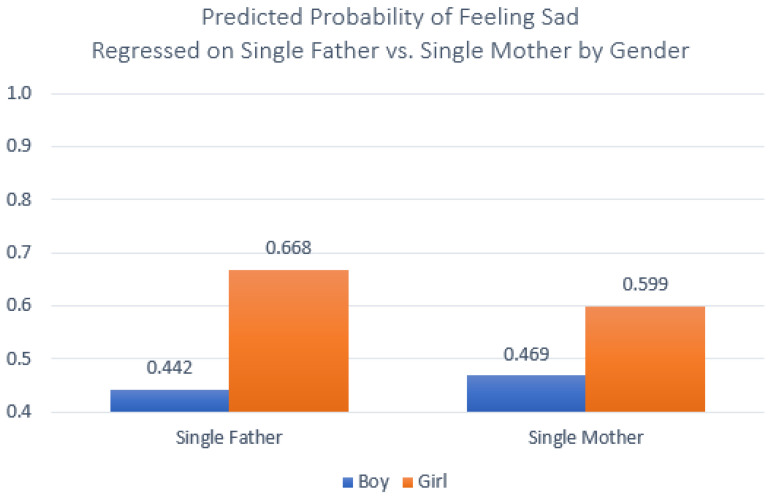
Interaction effect between single-father/single-mother status and child sex predicting odds of child feeling sad. Note: Data source is the 2021 Korea Youth Risk Behavior Survey. Single father *n* = 897; single mother *n* = 2751.

**Table 1 ijerph-20-03656-t001:** Description of all the variables in the model for the total sample and by family structure.

	Total (*n* = 38,383)					Both Parents (*n* = 34,735)					Single Father (*n* = 897)					Single Mothers (*n* = 751)				
	Freq.	% or Mean	SD	Min	Max	Freq.	% or Mean	SD	Min	Max	Freq.	% or Mean	SD	Min	Max	Freq.	% or Mean	SD	Min	Max
Family Structure																				
Both parents	34,735	0.91		0	1	34,735														
Single father	897	0.02		0	1						897									
Single mother	2751	0.07		0	1											2751				
How much felt stressed, usually (1 = not at all to 5 = very much)		3.27	0.95	1	5		3.26	0.94	1	5		3.41	0.93	1	5		3.38	0.97	1	5
Whether they felt extremely sad or frustrated in the past 12 months (0 = no; 1 = yes)		0.26		0	1		0.25		0	1		0.35		0	1		0.32		0	1
How often they felt lonely in the past 12 months (1 = hardly through 5 = always		2.51	1.05	1	5		2.49	1.04	1	5		2.73	1.07	1	5		2.69	1.08	1	5
Grade (7–12)		9.32	1.72	7	12		9.29	1.72	7	12		9.48	1.72	7	12		9.55	1.72	7	12
Boys	18,348	0.48		0	1	16,667	0.48		0	1	440	0.50		0	1	1224	0.44		0	1
Girls	20,035	0.52		0	1	18,068	0.52		0	1	457	0.50		0	1	1527	0.56		0	1
Self-reported school performance (1 = low to 5 = high)	18,348	0.48		0	1		3.17	1.14	1	5		2.82	0.83	1	4		2.87	1.17	1	5
Self-reported socioeconomic status (1 = low to 5 = high	20,035	0.52		0	1		3.46	0.82	1	5		3.08	0.84	1	4		3.21	0.89	1	5
Father’s education (1 = less than middle school to 4 = college or more		2.97	0.65	1	4		2.94	0.64	1	4		2.82	0.83	1	4		3.04	0.77	1	4
Mother’s education (1 = less than middle school to 4 = college or more		2.89	0.65	1	4		2.90	0.64	1	4		3.08	0.84	1	4		2.82	0.72	1	4
Self-rated health (1 = poor to 5 = excellent)		3.77	0.90	1	5		3.78	0.89	1	5		3.57	0.93	1	5		3.66	0.94	1	5
Perceived Body Image																				
Too skinny	1619	0.04		0	1	1473	0.04				41	0.04				105	0.04			
Normal weight	34,051	0.89		0	1	30,912	0.89				746	0.84				2393	0.88			
Too fat	2713	0.07		0	1	2350	0.06				110	0.12				253	0.09			
Locality																				
Rural	2136	0.04		0	1	1900	0.04				77	0.07				159	0.05			
Small city	34,136	0.46		0	1	15,282	.46				434	0.48				1246	0.46			
Big city	19,285	0.50		0	1	17,553	0.50				386	0.45				1346	0.49			

Notes: Frequencies are unweighted, and % or mean are weighted. Data source is the 2021 Korea Youth Risk Behavior Survey; total *n*= 38,383 respondents (both parents *n* = 34,735; single father *n* = 897; single mother *n* = 2751).

**Table 2 ijerph-20-03656-t002:** Ordinary least squares regression of child stress on family structure, demographic characteristics, resources, child self-perception and health, and other controls.

	Model 1			Model 2			Model 3			Model 4		
	Coeff.	S.E.		Coeff.	S.E.		Coeff.	S.E.		Coeff.	S.E.	
Constant	3.262	0.001	***	3.098	0.007		4.202	0.007		4.206	0.007	***
Family Structure (ref. Single Father)												
Single father	0.132	0.005	***	0.094	0.005	***	0.044	0.005	***	−0.016	0.007	***
Single mother	0.121	0.003	***	0.068	0.003	***	0.057	0.003	***	0.022	0.004	***
Grade (7 to 12)				0.035	0.000	***	0.028	0		0.028	0	***
Sex (ref. Boy)												
Girl				0.295	0.001	***	0.233	0.001		0.226	0.001	***
Socioeconomic status (1 = low to 5 = high)				−0.072	0.001	***	−0.02	0.001	***	−0.02	0.001	***
Father’s education (1 to 4)				−0.006	0.001	***	−0.014	0.001	***	−0.014	0.001	***
Mother’s education (1 to 4)				−0.018	0.001	***	−0.023	0.001	***	−0.022	0.001	***
School performance (1 = low to 5 = high)							−0.01	0.001	***	−0.011	0.001	***
Perceived Body Image (ref. Normal)												
Too skinny							0.002	0.003		0.003	0.003	
Too fat							0.082	0.003	***	0.082	0.003	***
Self-rated health (1 = poor to 5 = excellent)							−0.293	0.001	***	−0.293	0.001	***
Locality (ref. Big City)												
Rural area							−0.023	0.004	***	−0.024	0.004	***
Small city							−0.018	0.001	***	−0.018	0.001	***
Interaction												
Single father × Girl										0.117	0.009	***
Single mother × Girl										0.063	0.005	***
R Square	0.001			0.037			0.114			0.114		

Notes: Model 1 includes family structure (FS); Model 2 includes FS and demographic characteristics of child age, child sex, family SES, and parental education; Model 3 includes all variables in Model 2 plus child school performance, child perceived body image, child self-perceived health, and rurality. Model 4 includes all variables in Model 3 and adds interaction effects for family structure and child sex. Data source is the 2021 Korea Youth Risk Behavior Survey; total *n*= 38,383 respondents (both parents *n* = 34,735; single father *n* = 897; single mother *n* = 2751); *** *p* < 0.001.

**Table 3 ijerph-20-03656-t003:** Ordinary least squares regression of child loneliness on family structure, demographic characteristics, resources, child self-perception and health, and other controls.

	Model 1			Model 2			Model 3			Model 4		
	Coeff.	S.E.		Coeff.	S.E.		Coeff.	S.E.		Coeff.	S.E.	
Constant	2.492	0.001	***	2.898	0.007	***	3.876	0.008	***	3.880	0.008	***
Family Structure (ref. Single Father)												
Single father	0.239	0.005	***	0.195	0.005	***	0.151	0.005	***	0.109	0.007	***
Single mother	0.185	0.003	***	0.123	0.003	***	0.114	0.003	***	0.054	0.004	***
Grade (7 to 12)				0.005	0.000	***	−0.002	0.000	***	−0.002	0.000	***
Sex (ref. Boy)												
Girl				0.284	0.002	***	0.228	0.001	***	0.219	0.002	***
Socioeconomic status (1 = low to 5 = high)				−0.114	0.001	***	−0.066	0.001	***	−0.066	0.001	***
Father’s education (1 to 4)				−0.025	0.001	***	−0.032	0.001	***	−0.032	0.001	***
Mother’s education (1 to 4)				−0.047	0.001	***	−0.051	0.001	***	−0.050	0.001	***
School performance (1 = low to 5 = high)							−0.018	0.001	***	−0.019	0.001	***
Perceived Body Image (ref. Normal)												
Too skinny							−0.011	0.004	**	−0.010	0.004	**
Too fat							−0.013	0.003	***	−0.012	0.003	***
Self-rated health (1 = poor to 5 = excellent)							−0.256	0.001	***	−0.256	0.001	***
Locality (ref. Big City)												
Rural area							−0.026	0.004	***	−0.027	0.004	***
Small city							0.021	0.001	***	0.021	0.001	***
Interaction												
Single father × Girl										0.082	0.010	***
Single mother × Girl										0.108	0.006	***
R Square	0.003			0.034			0.081			0.082		

Notes: Model 1 includes family structure (FS); Model 2 includes FS and demographic characteristics of child age, child sex, family SES, and parental education; Model 3 includes all variables in Model 2 plus child school performance, child perceived body image, child self-perceived health, and rurality. Model 4 includes all variables in Model 3 and adds interaction effects for family structure and child sex. Data source is the 2021 Korea Youth Risk Behavior Survey; total *n* = 38,383 respondents (both parents *n* = 34,735; single father *n* = 897; single mother *n* = 2751); ** *p* < 0.01. *** *p* < 0.001.

**Table 4 ijerph-20-03656-t004:** Logistic regression of child sadness in the past 12 months on family structure, demographic characteristics, resources, child self-perception and health, and other controls.

	Model 1			Model 2			Model 3			Model 4		
	Coeff.	O.R.		Coeff.	O.R.		Coeff.	O.R.		Coeff.	O.R.	
Constant	−1.072	0.342	***	−0.840	0.432	***	0.895	2.447	***	0.914	2.495	***
Family Structure (ref. Single Father)												
Single father	0.441	1.554	***	0.415	1.515	***	0.316	1.372	***	−0.003	0.997	
Single mother	0.304	1.355	***	0.244	1.276	***	0.222	1.249	***	0.146	1.157	
Grade (7 to 12)				0.022	1.022	***	0.002	1.002	†	0.002	1.002	
Sex (ref. Boy)												
Girl				0.441	1.554	***	0.374	1.454	***	0.350	1.419	***
Socioeconomic status (1 = low to 5 = high)				−0.095	0.909	***	0.021	1.021	***	0.021	1.021	†
Father’s education (1 to 4)				−0.050	0.952	***	−0.068	0.934	***	−0.068	0.934	***
Mother’s education (1 to 4)				−0.071	0.932	***	−0.075	0.928	***	−0.074	0.928	***
School performance (1 = low to 5 = high)							−0.144	0.866	***	−0.144	0.866	***
Perceived Body Image (ref. Normal)												
Too skinny							0.032	1.032	***	0.032	1.033	***
Too fat							0.013	1.013	*	0.012	1.012	***
Self-Rated Health (1 = poor to 5 = excellent)							−0.384	0.681	***	−0.385	0.681	***
Locality (ref. Big City)												
Rural area							0.032	1.032	***	0.030	1.031	†
Small city							0.041	1.041	***	0.040	1.041	***
Interaction												
Single father × Girl										0.582	1.790	***
Single mother × Girl										0.127	1.136	***
−2LL	2,115,969.958			2,092,479.257			2,041,495.046			2,040,745.449		

Notes: Model 1 includes family structure (FS); Model 2 includes FS and demographic characteristics of child age, child sex, family SES, and parental education; Model 3 includes all variables in Model 2 plus child school performance, child perceived body image, child self-perceived health, and rurality. Model 4 includes all variables in Model 3 and adds interaction effects for family structure and child sex. Data source is the 2021 Korea Youth Risk Behavior Survey; total *n* = 38,383 respondents (both parents *n* = 34,735; single father *n* = 897; single mother *n* = 2751).; † *p* < 0.1. * *p* < 0.05. *** *p* < 0.001.

**Table 5 ijerph-20-03656-t005:** Ordinary least squares regression of child stress on single father or single mother status, demographic characteristics, resources, child self-perception and health, and other controls.

Family Structure (Ref. Single Father)												
Single Mother	−0.011	0.006	*	−0.052	0.006	***	−0.016	0.005	***	0.004	0.008	
Grade (7 to 12)				0.026	0.001	***	0.034	0.001	***	0.034	0.001	***
Sex (ref. Boy)												
Girl				0.344	0.005	***	0.301	0.005	***	0.331	0.009	***
Socioeconomic status (1 = low to 5 = high)				−0.06	0.003	***	−0.024	0.003	***	−0.023	0.003	***
Father’s education (1 to 4)				0.039	0.003	***	0.035	0.003	***	0.035	0.003	***
Mother’s education (1 to 4)				−0.076	0.004	***	−0.068	0.003	***	−0.067	0.003	***
School performance (1 = low to 5 = high)							0.013	0.002	***	0.013	0.002	***
Perceived Body Image (ref. Normal)												
Too skinny							0.021	0.012	†	0.02	0.012	†
Too fat							0.154	0.008	***	0.153	0.008	***
Self-rated health (1 = poor to 5 = excellent)							−0.264	0.002	***	−0.264	0.002	***
Locality (ref. Big City)												
Rural area							0.028	0.012	*	0.027	0.012	*
Small city							0.017	0.005	***	0.017	0.005	***
Interaction												
Single mother × Girl										−0.039	0.011	***
R Square	0.000			0.045			0.115			0.115		

Notes: Model 1 includes single-father/single-mother status (SFSM); Model 2 includes SFSM and demographic characteristics of child age, child sex, family SES, and parental education; Model 3 includes all variables in Model 2 plus child school performance, child perceived body image, child self-perceived health, and rurality. Model 4 includes all variables in Model 3 and adds interaction effects for family structure and child sex. Data source is the 2021 Korea Youth Risk Behavior Survey. Single father *n* = 897; single mother *n* = 2751.; † *p* < 0.1. * *p* < 0.05. *** *p* < 0.001.

**Table 6 ijerph-20-03656-t006:** Ordinary least squares regression of child loneliness on single father or single mother status, demographic characteristics, resources, child self-perception and health, and other controls.

	Model 1			Model 2			Model 3			Model 4		
	Coeff.	S.E.		Coeff.	S.E.		Coeff.	S.E.		Coeff.	S.E.	
Constant	2.731	0.005	***	2.874	0.024	***	3.540	0.026	***	3.548	0.026	***
Family Structure (ref. Single Father)												
Single mother	−0.054	0.006	***	−0.088	0.006	***	−0.052	0.006	***	−0.067	0.009	***
Grade (7 to 12)				−0.004	0.002	**	0.001	0.002		0.001	0.002	
Sex (ref. Boy)												
Girl				0.372	0.005	***	0.337	0.005	***	0.315	0.010	***
Socioeconomic status (1 = low to 5 = high)				−0.056	0.003	***	−0.019	0.003	***	−0.020	0.003	***
Father’s education (1 to 4)				0.016	0.004	***	0.009	0.004	*	0.009	0.004	*
Mother’s education (1 to 4)				−0.054	0.004	***	−0.047	0.004	***	−0.047	0.004	***
School performance (1 = low to 5 = high)							−0.015	0.002	***	−0.014	0.002	***
Perceived Body Image (ref. Normal)												
Too skinny							0.068	0.014	***	0.068	0.014	***
Too fat							0.036	0.009	***	0.037	0.009	***
Self-rated health (1 = poor to 5 = excellent)							−0.229	0.003	***	−0.229	0.003	***
Locality (ref. Big City)												
Rural area							−0.001	0.013		−0.001	0.013	
Small city							0.067	0.005	***	0.067	0.005	***
Interaction												
Single mother × Girl										0.028	0.012	*
R Square	0.000			0.036			0.078			0.078		

Notes: Model 1 includes single-father/single-mother status (SFSM); Model 2 includes SFSM and demographic characteristics of child age, child sex, family SES, and parental education; Model 3 includes all variables in Model 2 plus child school performance, child perceived body image, child self-perceived health, and rurality. Model 4 includes all variables in Model 3 and adds interaction effects for family structure and child sex. Data source is the 2021 Korea Youth Risk Behavior Survey. Single father *n* = 897; single mother *n* = 2751.; * *p* < 0.05. ** *p* < 0.01. *** *p* < 0.001.

**Table 7 ijerph-20-03656-t007:** Logistic regression of child sadness in past 12 months on single-father or single-mother status, demographic characteristics, resources, child self-perception and health, and other controls.

	Model 1			Model 2			Model 3			Model 4		
	Coeff.	O.R.		Coeff.	O.R.		Coeff.	O.R.		Coeff.	O.R.	
Constant	−0.632	0.532	***	−1.234	0.291	***	−0.099	0.906	†	−0.232	0.793	***
Family Structure (ref. Single Father)												
Single mother	−0.136	0.872	***	−0.206	0.814	***	−0.124	0.883	***	0.107	1.113	***
Grade (7 to 12)				0.073	1.075	***	0.074	1.077	***	0.072	1.075	***
Sex (ref. Boy)												
Girl				0.623	1.864	***	0.627	1.872	***	0.93	2.534	***
Socioeconomic status (1 = low to 5 = high)				−0.098	0.907	***	−0.014	0.986	*	−0.012	0.988	†
Father’s education (1 to 4)				0.046	1.047	***	0.018	1.018	*	0.019	1.019	*
Mother’s education (1 to 4)				−0.083	0.92	***	−0.078	0.925	***	−0.079	0.924	***
School performance (1 = low to 5 = high)							−0.145	0.865	***	−0.149	0.862	***
Perceived Body Image (ref. Normal)												
Too skinny							0.361	1.434	***	0.353	1.424	***
Too fat							0.324	1.382	***	0.307	1.359	***
Self-Rated Health (1 = poor to 5 = excellent)							−0.296	0.744	***	−0.3	0.741	***
Locality (ref. Big City)												
Rural area							−0.01	0.99		−0.011	0.99	
Small city							0.039	1.04	***	0.04	1.041	***
Interaction												
Single mother × Girl										−0.405	0.667	***
−2LL	208,394.843			203,740.853			199,046.701			198,796.564		

Notes: Model 1 includes single-father/single-mother status (SFSM); Model 2 includes SFSM and demographic characteristics of child age, child sex, family SES, and parental education; Model 3 includes all variables in Model 2 plus child school performance, child perceived body image, child self-perceived health, and rurality. Model 4 includes all variables in Model 3 and adds interaction effects for family structure and child sex. Data source is the 2021 Korea Youth Risk Behavior Survey. Single father *n* = 897; single mother *n* = 2751.; † *p* < 0.1. * *p* < 0.05. *** *p* < 0.001.

## Data Availability

Korea Youth Risk Behavior Survey (KYRBS) data are publicly available and can be acquired from the website at https://www.kdca.go.kr/yhs/home.jsp, accessed on 30 August 2022.

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
