# Peer review of "Associations between Gendered Family Structures and Adolescent Stress, Loneliness, and Sadness in South Korea"

_ijerph, 2023, doi:10.3390/ijerph20043656_

Round 1
Reviewer 1 Report
I am very strongly biased in this topic due to my own empiric and anecdotal experience as well as evidence in Europe (Germany and Romania) and Japan. The paper is relevant, well-written and of great interest to readers - nevertheless, I have some observations which might be included in further research or contribute to a more nuanced conclusion in this one:
1. Given South Korea's extremely low birth-rate (the lowest in the world if I am informed correctly and far below the historically acknowledged replacement rate), why do such papers which strive to reinforce the traditional gender distribution are still published? I mean, if things were so great in "traditional two-parents family" systems, then children born and educated in such system would happily replicate them, wouldn't they?
2. The observation that the wealthier become, the less children they (choose to) bring into the world stands in South Korea's case, as it does in Japan's, China's and many of Western or other nations: nevertheless, according to various reports over the past several years, there is a clear correlation between the world's "happiest nations with happiest citizens", the world's nations with the least - if any, at all - gender gap as well as gender gap in retirement pensions, and the world's nations with most children born out of wedlock - all of them being the so-called Northern or Scandinavian countries. It is a mystery to me every time I read such scientific papers as the current one - all of the with extremely well documented resources and research - which carry in the background conservative assumptions of "traditional two-member of the opposite gender family" system as being better than its alternatives ... well, present-day global data show otherwise and time might have come, even for Asian-oriented research, to ask the hard questions as to why the traditional family structures do not - obviously - work despite delivering positive sociological results and then slowly, gradually to shift the focus from proving by all means, scientifically accurate in itself but wrong in the larger scheme of things, that traditional family structures are right over alternative ones towards historical reality as a barometer for socioeconomically relevant and scientifically accurate research. Study like this are employed by socioeconomic factors and policymakers to reject any change and therefore to continue the, clearly, historically detrimental policies and strategies. as well as for the population to keep on discriminating children born and/or raised outside what is considered socially accepted norm.

Reviewer 2 Report
General comments: Thank you so much for giving me the opportunity to review this well written and structured manuscript. The authors have clearly provided detailed methods of the study procedures with adequate data interpretation. Additionally, they have explicitly demonstrated the limitations of their study with clear directions for future research. However, I have some comments that would help in improving this manuscript.
There were some grammatical errors in the manuscript that I would recommend the authors to review.
Introduction:
It is well written, providing clear, sufficient background information but I just had few comments.
In line 35, I suggest removing such families
In Line 38: I suggest removing “than in western countries” as it is repeated late on in the sentence
In line 40: I suggest removing “for”
In lien 46: put “increase” instead of “increases”
Methods:
In Line 224: “contains” should be “contain”
Table 1: I suggest adjusting the headings in the first row so they could be easily read and also the numbers within the table need to be adjusted so they are placed under the appropriate headings.
Analytic plan: I would suggest adding here the statistical software used in the analysis.
Results and discussion: These sections are clearly written and described in detail.
For future research, the current study evaluated stress, loneliness and sadness through single question for each which might not provide detailed information about each of these variables, so I suggest adding this point in in the study limitations and highlighting the importance of looking at these variables in detail in future studies.
Reviewer 3 Report
This is an important article examining the effects of single-mother families, single-father families, and two-parent families on adolescents. It is a strength of this work to examine children's mental health rather than relying exclusively on school performance as a measure. The focus upon stress, loneliness, and sadness are very important variables. The article is very well-written although long and looking at a complex range of issues.
The literature review is very good. The section that addresses whether "mothering" and "fathering" are inherently different is especially strong.
The huge sample of almost 40,000 adolescents is another important strength of this article. The up-to-date data from the Korea Youth Risk Behavior Survey of 2021ensures the relevance of this work.
The description of the methods utilized is strong. I am not a research methodologist and hope another reviewer will be helpful here. The bar graphs nicely illustrate the important findings and make findings accessible to readers who do not have statistical expertise and may find the 4 models inaccessible.
The reader learns the differences between the constructivist perspective and the essentialist perspective, with evidence tending toward essentialist theory.
The Discussion section is strong. Overall, the reader learns that two-parent families have advantages over both single-mother families and single-father families. And that adolescents in single-father families experience more difficulties, especially girls. The description of the limitations of the work is good. The authors do a very good job in suggestions for future research.
There is only one concern I have regarding this work that I feel the authors need to address. What about siblings? It is unclear whether some of the adolescents have siblings and what impact siblings have on the results of this work? Clearly, stress, loneliness, and sadness have some relationship to the actual family constellation. These three variables may be related to birth order. For example, the oldest sibling may experience heightened feelings of stress, loneliness, and sadness in having to take on parental roles. On the other hand, having siblings can help to reduce these feelings. I do hope the authors will be able to offer some explanation.
Glaring omission is the absence or presence of siblings
